© Author(s) 2017. CC BY 4.0 License.

### A DECISION SUPPORT SYSTEM (DSS) FOR CRITICAL LANDSLIDES AND ROCKFALLS AND ITS APPLICATION TO SOME CASES IN THE WESTERN ITALIAN ALPS

Davide Bertolo<sup>1</sup>

<sup>1</sup>Struttura organizzativa attività geologiche, Regione Autonoma Valle d'Aosta, Quart (AO), 11020, Italy. *Correspondence to*: Davide Bertolo (d.bertolo@regione.vda.it)

#### Abstract

Operative geologists who are involved in emergency management have often to deal with the 10 consequences of assuming critical and strongly impacting decisions in uncertain conditions.

Geohazards induced by active landslides are one of the civil protection situations requiring such decisions.

Nowadays, the monitoring of active landslides is almost always supported by numerical early warning systems, based on instrumental geotechnical and topographic networks. These networks provide

numerical early warning thresholds, which are set up in order to activate alert conditions at various levels of criticality in an objective way.
Desire the set of the set of

Despite these progresses the issue related to the possibility to dispatch false alerts has not yet effectively solved and that's the reason why the critical stages of the decisional processes are frequently relying not only on quantitative thresholds but also on the subjective experience of the emergency managers.

Therefore it is not so uncommon to read landslide-monitoring procedures that combine the quantitative information provided by the monitoring systems with the qualitative decisional elements coming from their professional experience in order to assume the most correct decision.

It's therefore evident that such an approach weakens the objectiveness provided by instrumental monitoring systems but, at the same time, collecting geological empirical and qualitative data can strengthen an hypothesis like the one that an active landslide could finally collapse.

Bayesian methods are frequently used in clinical decision making, another field of the human activity where critical decisions have to be made in a short time, combining objective values such as those provided by medical tests with diagnostic qualitative markers.

Based on the methods of clinical diagnosis, the has author has elaborated a reliable and objective
Bayesian Decision Support System (or DSS), developed to support the decision makers in assuming the most correct decisions based on all the elements, both quantitative and qualitative, that are available at a certain step of the decision process.

Thanks to the Bayesian approach, the DSS allows also to assess the predictivity of any single decisional step, which is the probability that a monitored landslide actually collapses when particular diagnostic

evidences are detected, either instrumental or observational. Hence the decision makers who are able to issue a civil protection alert when a given degree of confidence about the chance that a monitored landslide will collapse is reached. The degree of

confidence associated to the civil protection alert can be declared in the alert bulletin (e.g.: 80% or 93%). The decisional process can be tracked and replied by everyone in complete transparency.

It's therefore evident that such a DSS allows the civil protection authorities to increase the reliability of the alerts, reducing at the same time the so-called "cry wolf" effect and the discomfort related to

evacuations and to other civil protection measures. As a matter of fact, the decisional process becomes clearer and the people's trust in the civil protection systems is being strengthened by a more transparent emergency communication.

The DSS here described is an evolution and a statistical improvement of the method adopted in 2013 and 2014 during the emergency of the Mont the la Saxe landslide, and is now being successfully applied

to two other hazardous situations in the Aosta Valley Alps: the Brenva Site (Mont Blanc Massif) and the Berlachu site in the municipality of Lillianes (Lower Lys Valley).

#### 1. The DSS fundamentals

The rationales at the basis of the numerical methods for landslide monitoring are well known and are adopted by nearly every operator in the field of the natural hazard managing.

- 15 During the last years the numerical monitoring has gained an unquestionable supremacy on all the other "qualitative" monitoring and surveying methods and enhanced by the developments in the accuracy of the remote and contact monitoring instrumentations (Terzaghi, 1950; Dunnicliff, 1994). Furthermore, during the last decades the increasing availability of numerical data has supported the development of empirical and statistical methods, in the effort to implement reliable landslide early
- 20 warning systems based on objective thresholds (Voight, 1988; Voight, 1989). Today, these statistical methods, widely adopted for landslide monitoring, can't be successfully used without being placed within a larger and harmonic logical framework, which should be aimed to correctly manage hazardous and emergency situations.

In this light, it has become necessary to have a global framework including allowing a comprehensive

approach to these kind of phenomena (Crosta et al., 2012), including also an analysis of the targets at stake, the civil protection measures adopted to protect human lives and the economic assets, the impacts coming from the enforcement of the civil protection measures and an effective risk communication system.

The assumptions at the basis of the method proposed in this paper rely on some famous studies about 30 human behavior in decisional processes and the cognitive biases which influence these activities (Kahneman and Tvertsky, 1979).

The two authors demonstrated that human behavior, facing decisions implying potential losses, is usually biased by the so-called "risk aversion" mechanism: the human brain, facing risky choices, tends to emphasize losses and to choose conservative solutions.

On the other hand, when the so-called "overconfidence" prevails in judgments, often sustained by a consistent professional experience, this is the kind of behavior that can easily lead to catastrophic effects (Pulford and Colman, 1996; Kahneman D., 2011). Other biases, like the "confirmation bias" will be also examined as factors potentially influencing a decisional process.

As previously said, when monitoring active landslides, one way to overcome these biases has been, so 40 far, the adoption of numerical models based on instruments able to detect even the smaller

© Author(s) 2017. CC BY 4.0 License.

displacements and to implement early warning DSS exclusively based on these instruments and warning thresholds.

Nevertheless, the overconfidence about the capacity of a completely automatized system to be absolutely reliable without any human intervention could lead to a substantial underestimation of

qualitative evidences.

Such evidences, which are the so-called "expert judgments", are usually based on the observation of terrain evidences, which are equally worth to be weighted in the decision process. Regarding large landslides, among these evidences are well known qualitative signs such as the appearance of fractures, smaller rockfall frequency increase, and so on.

- 10 When applied to a real civil protection emergency situation, the mixing of the two kind of data often produce quite aleatory results, because in several EW procedures the early warning decisional stages based on automatic monitoring networks are followed by further and not well defined "advices from experts" or "geological field investigations", to be acquired once the EWS quantitative values have been exceeded in order to support the issuing of the alerts.
- 15 It is therefore obvious that, when expert judgments are required to support decisions in the advanced stages of the EW procedures, the opposite biases of risk aversion or overconfidence will probably shift the expert judgments, made by humans, either to confirm the numerical values (risk aversion) or, even worse, to refute or underestimate the numerical results coming from the EWS networks under the overconfidence bias.
- To further complicate that framework, here comes today the increasing availability of a great amount of 20 data provided by the monitoring networks, particularly when an instability is continuously monitored by different redundant networks at an high-frequency sampling rate.

As it will be shown later, the implementation of a Bayesian DSS of the type here described, contributes to establish a "network hierarchy" will make also easy to assess which are the best monitoring systems

in order to provide the decision-makers the best monitoring technologies to support their evaluations. The DSS has been implemented by the geological survey (Struttura attività geologiche) of the Valle d'Aosta Autonomous Region and is being successfully tested at three test sites on the regional territory, with different kinds of slope instability.

#### 1.1 The Mont de La Saxe landslide experience 2012-2014

In order to correctly explain the bases of the proposed method, the experience made by the regional geological survey in monitoring the evolution of the Mont de La Saxe landslide since 2012 will be described.

The Mont de La Saxe Landslide is one of the most active and dangerous large slope instabilities in the Alps. According to current estimates, it involves an area of about 150.000 m<sup>2</sup>, with a maximum width

of 350 m and a maximum length of 500 m. The total mass potentially involved in a collapse is about 8,0  $\div$  8.5 \*10<sup>6</sup> m<sup>3</sup>. The potential collapse threatens an important touristic resort in the western Italian Alps and important infrastructures such as the Mont Blanc Tunnel and the E25 Motorway which are among the most important road links between the industrial areas of northwestern Italy, France and Switzerland. (Fig. 1)

The importance of the targets at stake led to the implementation of a very redundant and performing monitoring EW system, active since 2009.

The first EW procedure, elaborated in 2008, followed the classical and common approach to the alert dispatching system, based on the exceeding of two velocity thresholds, respectively of V<sub>24</sub>=1mm/h

(pre-alarm condition) and  $V_{24}=2$ mm/h (alarm condition). The two thresholds are calculated measuring hourly the surface slope deformation by a robotized total station (RTS) on about 30 optical targets placed on the unstable slope. The  $V_{24}$  values come from the calculation of the mean of the hourly measurements on a 24-hour time interval.

The displacement thresholds are calculated applying the Voight model (Voight, 1989; Tamburini and

10 Martelli, 2006). The overcoming of the alert threshold of  $V_{24}=2mm/h$  required the dispatching of an alert message to the civil protection authorities and should be followed by a quite unclear "consulting phase" in which experts had to be consulted and further investigations should be undertaken.

On may 5<sup>a</sup>, 2012, the EW network detected the first exceeding of a warning threshold since its activation in 2009. The following field investigations showed that a cinematic-structural domain of the

15 monitored landslide, involving a volume of about 650.000 m<sup>3</sup> (C domain in Fig 2.) was becoming unstable. V24

After about 30 days, the displacement velocity of the C domain returned below the threshold of  $V_{24}$ = 2mm/h and the alarm condition was revoked. Terrain evidences like neo-formation scarps and fractures appeared during the summer 2012, as a sign of an in-progress evolutionary process.

- In 2013, at the end of March, the combination of a thick snow cover (about 2000 mm cumulated snow cover), suddenly melted by a warm föhn wave, triggered a massive activation of the unstable domain. The destabilization of the C domain was followed by the activation on the entire unstable slope, due to the "recall" effect of the neighboring sectors. The risk of a multiple sequential failure potentially leading to the collapse of the whole landslide became real.
- Consequently, from 19/04/2013 to 24/06/2013 the massive activation forced the geological survey to 25 issue a new warning. An estimated 650.000 m<sup>3</sup> volume could collapse, potentially followed by the collapse of the entire unstable slope. The municipality of Courmayeur evacuated about 100 people and the access to the Val Ferret was interrupted. The emergency condition lasted for about 40 days and the alert thresholds set at 2 mm/h were uninterruptedly surpassed for nearly two months (maximum value
- of  $V_{24}=7$  mm/h). Nevertheless no collapse happened.

These experiences made it clear that the  $V_{24}=2mm/h$  threshold was not compliant to the real behavior of the landslide. Therefore, the decisional process in the EW procedure was integrated with a series of further decisional elements (Bertolo, 2013), which will be named later as decisional values.

A further and consistent proof of the inadequacy of the single-threshold approach finally came when the 35 landslide displacement rate newly increased at the end of March 2014, reaching in the C domain displacements up to 500 mm/hour. The emergency reached its top events with two main collapses from the C sector on April  $17^{\text{th}}$  and April  $21^{\text{th}}$ , 2014, with two single collapses respectively of  $5 \times 10^3 \text{ m}^3$  and

 $3x10^4$  m<sup>3</sup> each one. The rest of the domain C slowly collapsed transforming with an earthflow rheology and not threatening the targets. At the end of June 2014, the whole C domain had collapsed. Thanks to a procedure involving multiple decisional elements, the E25 route was closed only 3 hours and people had been evacuated for about 40 days, even if the instability had become more critical than in 2013.

The emergency management during 2013 and 2014 landslide activation performed very well, providing a relevant outcome of best practices in many domains, such as risk communication (Giordan et al, 2014) and failure potential forecasting methods (Manconi and Giordan, 2015).

During these emergencies the EW procedure was amended adopting a flexible approach: the exceeding of the  $V_{24}$  threshold led to an intensified surveillance phase, in which other elements such as the  $V_{24}$ 

- increasing in surrounding sectors and the in-depth acceleration detected by multiparametric probes, where adopted as decisional values to support the final decision to dispatch the civil protection alerts. Thus, being able to adapt the alert dispatching with a "learn as you go" method (Peck, 1969) showed to be an efficient way, minimizing the impact on the population lifestyle and local economy by the civil protection plan enforcement, assuring at the same time an appropriate safety level.
- Despite that satisfactory experience, the method applied in 2013 and 2014, that integrated multiple decisional values, remained substantially a qualitative/quantitative mixed one, with a basically empirical and heuristic approach.

To achieve a stronger and comprehensive method to integrate, manage and capitalize all the potentialities given by an integrated monitoring system it was therefore necessary to develop an

- objective, mathematical method, being able to take profit from the successful decisional processes followed during the 2013 and 2014 emergencies. That need led to the development of a decision support system (DSS) suitable to dispatch alerts, with a clear probabilistic and replicable degree of confidence and that can be successfully used both on large landslides and on smaller medium-size slope instabilities.
- To implement the DSS, the Author has looked to the domains of the human activity where, like in the applied geology, qualitative and quantitative data have to be integrated in order to make decisions about critical issues. The results of that research showed that the medical diagnosis is one of the most prominent human activities where advanced studies and decisional protocols showing such features have been elaborated.
- Medical clinical diagnostic processes show a great similarity with EW operational geology: the need to integrate quantitative data produced by highly technological systems with qualitative data gathered with traditional methods, the difficulty in modeling the real situation due to the extremely complex setting of the subject under investigation, the lack of data coming for in-depth direct investigations due to their cost and terrain difficulties, and most of all, the need to assume decisions rapidly collecting all the
- available data to assume the correct decision. The problem in assessing the probability of slope collapses is almost quite the same of the medical clinic, when the assigned task is to find a method to assess the probability of the collapse of an unstable slope, given the appearance of a diagnostic signal, such as the exceeding of a given threshold (quantitative) or the appearance of trenches (qualitative).

## **1.2** Estimating the Predictive Positive Value of EW thresholds under an operative Bayesian approach

In medicine DSS assuming both objective test data (pathological approach) and expert observations (clinical approach) are by now given as consolidated and widely applied in diagnosing diseases and in 5 performing wide scale screenings (Dixon-Woods et al., 2005; Voils et al., 2009.)

- One of the most reliable mathematic-based approaches, providing the ability to integrate quantitative and qualitative data, is the Bayesian approach (Bayes, 1706). Under the Bayesian approach all the elements leading to a "highly confident" decision are ruled by the Bayes Theorem and by its applications to the evaluation and decision process.
- 10 It is well known that, since the mid '90s, Bayesian methods are being applied to geohazards and in EWS theoretical design too, emphasizing the opportunity to integrate in the decision processes different data coming from different sources, thus refining the models and the hazard forecasts (Medina-Cetina and Nadim, 2007; Michoud et al., 2013).

To implement a Bayesian DSS method suitable to landslides or landslides, it was assumed that each qualitative or quantitative evaluation, worth to be incerted in the DSS process, could be assimilated to a

qualitative or quantitative evaluation, worth to be inserted in the DSS process, could be assimilated to a diagnostic test providing what is called the "decisional value". A decisional value is any quantitative or qualitative evidence being able to support (or not) hypothesis of the final collapse of the whole unstable slope.

To show the usefulness of the Bayesian method, it may be used to assess the fallacy of a single-

threshold DSS in the case it is based only on the exceeding of the  $V_{24}=2$  mm/h threshold measured by an instrument like a RTS (the use of other instruments like GPS or a GbSAR will yield the same results).

The real case is that of the Mont the la Saxe landslide EWS procedure before its first amendment in 2013.

The early warning V<sub>24</sub>=2 mm/h threshold based on the Voight model is assumed as a test, or marker, which, if positive (the case in which V<sub>24</sub>=2 mm/h is exceeded), provides a decisional value leading to the immediate dispatching of the alert condition to the civil protection authorities. As said in the introduction, one of the main assumptions supporting the use of a Bayesian DSS is that it

can provide the decision makers the degree of confidence associated to any single decisional value 30 during the decisional process, that is called the Positive Predictive Value, or PPV (Rothman K. J., 2012).

The PPV expresses as degree of confidence about the likelihood of the original hypothesis on the basis of the elements available at a certain moment. If the PPV of a single decisional value will be too low, then it will be necessary to look for other elements supporting or not the original hypothesis, the so-

called Null Hypothesis or H<sub>e</sub> (Neyman and Pearson, 1933). If the PPV of a single decisional value will be too low, then it will be necessary to look for other elements supporting or not the original hypothesis, the so-called Null Hypothesis or H0 (Neyman and Pearson, cit.).

© Author(s) 2017. CC BY 4.0 License.

Therefore, in the case of an EWS based on surface displacement, calculating the PPV of the " $V_{24}$  test" means answering the following question: if the exceeding of  $V_{24}=2$  mm/h is being observed, what will be the probability to observe the whole collapse of the unstable slope?

Applying the Bayes theorem, the PPV of V24=2 mm/h threshold test will be:

[1] 
$$P(A|B) = \frac{P(B|A)P(A)}{P(B)} = H_a$$

Where and A and B are, respectively, the occurrence of the whole collapse event and the exceeding of the threshold (the observed datum). P(B|A) is therefore the conditional probability that, given the exceeding of the V<sub>24</sub>=2 mm/h threshold the monitored landslide actually collapse, that is the "Predictive Positive Value" of the test or, vice-versa, the likelihood of H<sub>o</sub> under a single test.

- To do so, three fundamental parameters influencing the predictive value of the  $V_{24}>2$  mm/h test have to be assessed:
  - 1. The Sensitivity (Se): the percentage of unstable slopes that, in a particular area (local, regional, or wider) at the time of their collapse, actually had exceeded V<sub>24</sub>=2mm/h;
  - 2. The Specificity (Sp): the percentage of active instabilities that, once exceeded the  $V_{24}$ =2mm/h

threshold, actually collapsed that is, in medical language, the "true positives". The inverse of this parameter, that is 1-Sp is the percentage of "false positives" that is the percentage of instabilities that, despite exceeding the V<sub>24</sub>=mm/h threshold didn't collapsed (Fig. 3);

- 3. The Prevalence (Pr): this the observed frequency of the active landslide phenomena at a certain time in a certain geographic domain. This parameter varies widely depending on the extension of
- the area examined (local, regional, continental, worldwide) thus it is very difficult to estimate. Therefore, the choice of the area on which the Pr is evaluated is critical because, since landslides occur in mountains or hill regions, a wide scale assessment of Pr, involving also plain regions, will probably lead to unlikely and low Pr ratios. Moreover, Pr may be also influenced by the environmental changes, like global warming or rainfall increase, hence the assessment of Pr should
- take into account not only the area in which it's calculated but even the time interval adopted for the calculation.

Coming to the calculation of the above parameters, it is almost immediate to realize that Se=100%, because every recorded catastrophic landslide at its collapse (terminal velocity) exceeds the V24=2mm/h threshold. By contrast, Sp is more difficult to assess, mainly due to the too short historical records of

instrumental monitored landslides available for that kind of statistical analysis, (Crosta and Agliardi, 30 2003; Mazzanti et al., 2015).

Applying the corrections required by the estimation of Pr, Pr, Se, the [1] becomes:

[2] 
$$PPV = \frac{Sensitivity * Prevalence}{Sensitivity * Prevalence + (1 - Specificity) * (1 - Prevalence)}$$

Conservatively, the following values will be assumed:

Sp= 0,80: that is in 80% of the cases the instabilities that exceeded  $V_{24}$ =2mm/h actually collapsed, while in the remaining 20% of the cases the instabilities, even if they exceeded the threshold, didn't collapse, thus 20% is the percentage of "false positives" detected by the test. The conservative nature of that choice is due to the small sample on which the  $V_{24}$  threshold has been estimated.

- Pr, as said before, is perhaps one of the hardest parameters to calculate, due to the difficulty to gather accurate estimates about the statistical incidence of the different types of landsides. On the other hand, estimating an high value of Pr will lead to an overestimation of the PPV of the test, which, all in all, is a conservative outcome, therefore functional to our demonstration and suitable to be included in a civil protection EW system without leading to risky decisions due to overconfidence.
- In this case Pr has been calculated on the basis of the Aosta Valley Autonomous Region landslide susceptivity calculated in the IFFI landslide phenomena inventory (Giardino and Ratto, 2007). According to the IFFI inventory the Aosta Valley Autonomous Region, with a total surface of 3.262 km<sup>2</sup>, is one of the Italian territories most affected by landslides. Pr has therefore been estimated as the ratio between unstable surface and total regional area, which is very high, ranging around 15%. This
- value appears to be consistent with the values calculated by examining other landslide inventories (Malamud et al. 2004).

Applying the values in the [2], the PPV is:

[3] 
$$PPV(V24mm) = \frac{1*0,15}{1*0,15+(1-0,80)*(1-0,15)} = 46,88\% \approx 47\%$$

- This result clearly shows that an EWS based only on a single surface displacement velocity will reach at its best a PPV of 47% (which means a 53% probability of dispatching a false alarm). And it's worth noting that this poor confidence value is reached with a conservative approach, assuming a high value of prevalence of the landslide type and assuming that 80% of the landslides that exceed the  $V_{24}$  threshold will actually collapse (Pr=80%).
- Sp is greatly influenced by the amplitude of the sample and all the studies to calculate the displacement velocity parameter adopt the back analysis method, meaning that these studies rely on the very small sample made by historical landslides. This small representativeness of the sample is another limit of the velocity-based thresholds, which becomes evident if it is uncritically adopted as an exclusive basis of an EWS.
- Another important factor influencing the PPV of the V<sub>24</sub> threshold comes from the intrinsic nature of the sample used to evaluate the threshold itself. The sample is, indeed, almost always based on back-analyses performed on unstable slopes that in most cases actually collapsed (Crosta and Agliardi, cit.). This circumstance greatly affects the value of Pr and leads to overestimating the "false positives", thus leading to predict the collapses of slopes that actually will not collapse at all.

#### 1.2.1. Examining a DSS exclusively based on the inverse velocity method

Another widely used approach to landslide monitoring is the inverse velocity method (Fukuzono, 1985). It has been recently re-elaborated and tested, in-parallel with the Voight method, during the Mont de La

Saxe emergency of 2014 (Manconi and Giordan 2015). In their work the two authors performed an estimation of the reliability (R) of this test, obviously assuming that the reliability of the forecast is poised to increase when approaching the Time of Failure or ToF.

If the inverse velocity method is embedded in a Bayesian DSS, assuming a conservative approach, even for the inverse velocity method, the Se of the test can be fixed at a value of Se=75% that, according to the cited two authors, can be considered as an high reliable value. Since the inverse velocity method increases its reliability (the PPV) approaching to the ToF and since civil protection plans have to be activated with the due advance in order to ensure public safety, there is a consistent risk that decision makers approach too much to the ToF in order to dispatch a reliable alert.

Nevertheless, even if the decision-makers decide to take the above-mentioned risk, once the threshold is exceeded, the PPV of the test will only be 41.3%, with a probability of false warnings of 58.7%. In fact, as suggested by Rose and Hungr (cit.), the method should not be used as stand-alone but should be sustained by further quantitative or qualitative observations.

The two examples clearly show that when it comes to implementing an EW and managing DSS 15 emergencies, the usual choice for implementing an EWS on critical landslides based on the evaluation of a single threshold could easily lead to a "cry wolf" effect or, in the worst case, an over-reliance on a single method of assessment could lead to human losses if the method chosen does not take into account the actual geological, mechanical behavior and the rheology of the monitored slope.

# 20 2. Implementation of the Bayesian DSS to manage some active landslides in Valle d'Aosta: The Mont de La Saxe landslide DSS.

One of the main goals in designing a new and comprehensive DSS to manage the La Saxe EWS was to deal with the necessity to integrate quantitative and qualitative data in a clear and replicable process, without running the risk to lose any vital information.

The second goal was to obtain a numerical value of the PPV of the dispatched alerts in order to provide an objective value of the probability of the worst case scenario reached at any evaluation stage during the evolution of the landslide, i.e the confidence that all evidences, both qualitative and quantitative, provided to support the likelihood of the collapse event:  $H_0$ .

The achievement of these results, with a transparent and replicable decisional process, would also have

an immediate and positive impact on the quality of the emergency communication to the population and to the other civil protection actors.

Due to the importance of the targets at stake, it was decided to design the main architecture of the DSS based on quantitative decisional values and to use the qualitative ones to only to strengthen the reliability of each single decisional step.

The overall architecture of the Bayesian inferential engine implemented at the Mont de La Saxe Landslide is shown in the conceptual model of Fig. 4 and the Table 1 reports the decisional values adopted.

The initial observed frequency (prevalence Pr) is set at 15%, according to the assumptions made in the paragraph 1.1. The power of the Bayesian engine comes from the progressive increase of the "relative"

prevalence of the phenomenon due to the relative increasing of Pr.

More in detail, an iterative process is performed, where the a posteriori Probability Density Function (or PDF) of the former step becomes the a priori PDF of the further test and the true value is progressively approached.

Note that some decisional values, the increasing of the water table in-depth level, are numerical parameters objectively measurable. Despite that, these parameters do not have a robust statistical record to allow the evaluation of a numerical threshold. Moreover, these parameters are too linked to the behavior of the single phenomenon on which they are measured, i.e.: they are too site-specific.

The evaluation of these parameters is therefore maintained as quantitative, although the observation of some of them is important to sustain the decision to raise the alert condition to an upper level.

- The Table 1, once transformed in a decisional tree (Fig. 5), shows more clearly the decisional process that allows the alert dispatching to the civil protection authorities. Hence, another useful feature of the Bayesian DSS is that the iterative process allows the prediction of "intermediate stage scenarios" in case the process should not verify the H<sub>0</sub>. Thus it is possible to use the intermediate scenarios as outcomes of the DSS in the case H<sub>0</sub> is not verified (Fig. 6). Note that H<sub>0</sub> has to
- be maintained as the null hypothesis during the whole decisional process. As a matter of fact, should H<sub>0</sub> be changed during the evaluation process, the application of [3] will have to be adjusted due to the different Sp and Se values.

#### 3. Application of the DSS to the Brenva site rock-avalanche (Mont Blanc Massif, Italy)

The above exposed DSS is now being applied on another site, the Brenva site. The site is a steep rocky wall located in the Mont Blanc Massif above the Brenva glacier on the Col Moore ridge southeastern flank (Fig. 7 and 8).

In this case the potential collapse of a large scale rockfall from the slope, could trigger a large ice-rock avalanche like the catastrophic one that happened in 1997 (Barla et al, 2000; Deline P., 2001; Deline P., 2000).

2009).

At this site, the implementation of an integrated monitoring system faces both environmental and logistical issues due to the site altitude (about 3.800 m a.s.l.), and low temperature conditions (T ranging down to -30 °C). In such conditions the Regional geological survey was forced to install a monitoring system made by remote monitoring networks integrated by accelerometers placed to detect potential collapse forerunners.

Under those constraints, due to the relative lack of quantitative data, it was necessary to implement a DSS based mostly on qualitative decisional values. (Tab. 2)

The  $H_{\circ}$  (top event) in this case is a massive rockfall collapse having a volume V> 1\*10<sup>s</sup> m<sup>3</sup> that could evolve as an ice-rock avalanche. Winter conditions with consistent snow cover are supposed to be a

35 fundamental element to trigger such a phenomenon. The DSS implemented is therefore limited to the situation where a consistent snow cover is present and the rockfall volume is greater than  $1*10^{\circ}$  m<sup>3</sup>.

The parameters conditioning the decisional values (Pr, Se, Sp) are quite difficult to assess, because ice rock avalanche phenomena are quite uncommon and significant statistical records describing these phenomena are lacking.

Natural Hazards and Earth System Sciences

For the first decisional step, where Pr is significant because it influences the Pr values of the further steps in the decisional chain, a Pr value = 15% could be accepted, according to the results by Malamud et al., (cit.) regarding the correlation between rockfalls frequency and their volume.

It should be noticed that, even with poor quantitative decisional values, a PPV of 85% is reached within 5 two decision steps inserting in the process quantitative but objective decisional values such as rockfall frequency increase and so on. Further observations and decisional values can be inserted into the DSS, in order to strengthen the reliability of the alerts. According to that, a recently activated local accelerometric sensing network will probably provide further decisional elements (Amitrano et al., 2010).

10

#### 4. Application of the DSS at the Lillianes site (Lys Valley)

Lillianes is a small municipality located in the Lys Valley, in the southeastern zone of the Valle d'Aosta. On April 19<sup>a</sup> 2017 a rockfall collapse of about 70 m<sup>3</sup>, hit a municipal road. The geological inspection of the detachment zone highlighted the presence of a large unstable rock mass about having a

15 volume of about 250 m<sup>3</sup>. The odds of further collapses led to the precautionary closure of the road, isolating some villages, which hosted farms and touristic activities.

The time required by the construction of a new safer road and the necessity