# Peer review of "© Author(s) 2017. CC BY 4.0 License."

_Natural Hazards and Earth System Sciences, 2017_

## Referee Comment (RC1) · Anonymous Referee #1 · 15 Jan 2018

Dear Editor,

Please find here below my review of the paper nhess-2017-396:

A DECISION SUPPORT SYSTEM (DSS) FOR CRITICAL LANDSLIDES AND ROCK-FALLS AND ITS APPLICATION TO SOME CASES IN THE WESTERN ITALIAN ALPS

By

Davide Bertolo

This paper presents the strategy of a DSS that have been experience in Aosta Valle. It shows the lessons acquired during different crisis period of the La Saxe landslide. The

goal of the paper is to objectivize the decisions, taking into account expert knowledge in addition to threshold values that can be updated within the DSS. The porposed solution include Bayesian approach based on clinical diagnosis approach. It also explain how this DSS is now applied to other instabilities.

General comments

The paper is interesting, because it present a real experience of EWS on a very high hazardous landslide. It is noteworthy that such a paper based on experience is valuable and must be published. Nevertheless, the paper suffer from a few problems:

- It is a bit too long and can be reduced.

- The paragraphs (new lines) are too short all along the text; several new lines must be cancelled.

- The abstract must shorten, and to be a real summary of the paper instead of arguments.

- The paper dos not include a real introduction. The section 1 must reworked accordingly. It means emphasis on what are the issue with monitoring and a presentation of where and how existing EWS and DSS exists. A few more references must be added.

- The section 1.1: can change to 2. What do we learn from La Saxe. AS La Saxe is the base of the thoughts, the system must be fully presented with a map of sensors and monitoring system.

- I am not sure that the example of Lilianes site and the Brenva rock avalanche necessitate sections, in my opinion the mention that it has been applied to other site with providing the value of Pr, Se, etc. is sufficient. It does not bring any new arguments in the discussion.

If these recommendations are followed, in my opinion, the paper will be a very interesting and important contribution.

Specific comments

- P2 line 13: Introduction

- P2 line 15: what does mean "numerical monitoring"?

- P2 line 20: Fukuzono references can be added.

- At the end of the introduction must mention La Saxe landslide where it has been developed.

- P4 line 4: add a reference to "dispatching system" or explain. The 24 of V24: what does it mean?

- P4 line 4: add a reference to "dispatching system" or explain. The 24 of V24: P4 line 16: V24?

- P4 line 4: add a reference to "dispatching system" or explain. The 24 of V24: P4 line 18: "neo-formation scars and fractures" can expressed in a different way.

- P4 line 4: add a reference to "dispatching system" or explain. The 24 of V24: P4 line 4: add a reference to "dispatching system" or explain. The 24 of V24: P5: lines 13-14: I do not follow!

- P4 line 4: add a reference to "dispatching system" or explain. The 24 of V24: P6 lines 3-18: part of this can be moved to introduction.

- P4 line 4: add a reference to "dispatching system" or explain. The 24 of V24: P4 line 4: add a reference to "dispatching system" or explain. The 24 of V24: P6: lines 34-38: = repetition.

- P4 line 4: add a reference to "dispatching system" or explain. The 24 of V24: P7 line 5: I am not used that formulation, but in my opinion "=H0" is not standard natation for the Null hypothesis.

- P4 line 4: add a reference to "dispatching system" or explain. The 24 of V24: P7: a

few words must be added concerning the link between formula 1 and 2. It is not very informative to use A and B, may be they can be replace explicitly and better explained.

- P4 line 4: add a reference to "dispatching system" or explain. The 24 of V24: P9 line 12: (cit.) is not used in NHESS format. In addition, Rose and Hungr are not cited before.

- P4 line 4: add a reference to "dispatching system" or explain. The 24 of V24: P16 line 15: you can add the reference to the recent landslide classification (geotechnical) by Hungr et al. (2014).

- P4 line 4: add a reference to "dispatching system" or explain. The 24 of V24: Figure 1 it can be improved: Italy is not visible (lines would be better).

- P4 line 4: add a reference to "dispatching system" or explain. The 24 of V24: Figure 2: too small characters.
* * *

---

## Author Comment (AC1) · 18 Jan 2018

Dear Editor, I must thank the referee for his work and his precious remarks. These advices are being taken in charge and the manuscript will be improved as suggested. Unclear points will be clarified as well. Best regards. Davide Bertolo
* * *

---

## Referee Comment (RC2) · Anonymous Referee #2 · 5 Feb 2018

The Manuscript deals with the issue of early warning systems for active slow moving landslides. A method is proposed for a "decision support system", claimed to be objective and reliable, which was applied to "some" (three) cases in the Italian Alps. I believe that the Manuscript is written in a rather confuse way, and that the above claims are not supported by the actual content of the text. Morevoer, it occasionally contains questionable statements.

Overall review of the Manuscript:

The Abstract of a scientific paper is supposed to concisely describe the framework, purpose, method, results and conclusions; the proposed Abstract hardly contains such

information. Then, from the beginning of the main text up to section 1.2, the Manuscript is written in a colloquial fashion and it does not introduce the general framework of the problem to put the proposed method into proper context. The description of the "decision support system" is given in a qualitative way, without motivating the adopted numerical values or, worse, by explicitly mentioning that values are "reasonable", without providing any insight into the consequences of using those particular values, or uncertainty, or any form of validation. At least these are not explained. The two following sections are supposedly devoted to the application of the same method in two different sites. It is not clear to me what is it that the Author learned from the first case, let alone that the information provided in the two additional sections does not provide any more insight. Thus, how can the Author state, in the first few lines of the Conclusions section, that the method was "successfully applied"? Where are evidences supporting that claim? The remaining of the Conclusions section fails to report actual "conclusions" of the effort made by the Author. What do we learn from reading the Manuscript? Why does the Author suggests not "installing useless and redundant instrumentation", where is evidence supporting that statement? The reason why would he like to revise the commonly accepted definition of landslides is also obscure to me. Eventually, the language throughout the text is sloppy; I am not an English native speaker myself, but I could spot several mistakes and, occasionally, repeated words and even a three-lines sentence (end of page 6) denoting careless writing.

Additional comments:

There are many expressions used in a confusing way. For example, the Author refers to "numerical methods" for landslide monitoring: "numerical methods" are usually devoted to computer-based problem solving, typically optimization or differential equation solving, which is not the case here. There are whole sentences that have little to do with the content of the paper (i.e. lines 29-38, page 2, about "human behaviour" and its supposedly relevant role in decisional processes), or the many references to "clinical practice", whose only overlap with the content of the Manuscript is the use of Bayes

theorem. There are a few confusing sentences bringing conceptually different issues to the same level, for example: i) in line 39, the Author refers to "monitoring" and "bias" on the same footing, while monitoring does not contain biases per se, being a quantitative measure; it may have different degrees of approximations or confidence, but surely it does not represent a bias; ii) in line 6, page 3, the sentence "Such evidences, which are the so-called expert judgements" contains the same kind of flaw: an "evidence" cannot be a "judgement". Similar and even worse problems are represented by the use of the word "objective" throughout the Manuscript. Objectively defining values for parameters in a model, such as the various thresholds and percentages values used here, is not equivalent to select them arbitrarily. Of course one can follow a trial and error procedure, but in that case one would also have to make sure if initail (the trial part) values were properly selected, or should they be modified (the error part). It seems that the decisions system's parameters were rather arbitrarily selected; if not, the Author failed to describe the method properly, in my opinion. And, by the way, "parameters" are contstant values in a mathematical model, which are used to represent the relationship between the "variables" of the model itself; parameters can be tuned, not measured (at variance with line 5, page 10). Variables are the measurable quantities, and bear physical meaning.

The use of Bayes theorem is actually my major concern, here. Bayesian inference is a statistical method, i.e. it is meant to estimate a probability density function, based on the concepts of prior and conditional probabilities, and on the idea that posterior probabilty can be updated whenever new evidence comes about. It is understood that new evidence must concern the SAME variables used in previous estimates. Otherwise, we start describing a different phenomenon, or a different model. It seems to me that in the process described by the Author each different step (cf. Fig. 5) brings about NEW variables, and no probability is defined for the overall set of variables to simultaneously assume different values (conditional probability), and no prior estimate of such variables to assume those values was defined (prior probability). If so, I do not see how can one claim to enforce Bayes theorem, or I am missing something fundamental here. This add to the issue of how the numerical values of the thresholds were selected, discussed above. Moreover, in line 32, page 7, what does "Applying the corrections required .. [1] becomes" mean? Correction required by whom? How is Eq. (2) implied by Eq. (1)?

In line 11, page 8, the Author refers to susceptivity (usually referred to as susceptilibity) as the ratio of unstable area to total area under investigation, which is a bit of a semplification, given the body of literature existing on the topic of landslide susceptibility assessment... Also, the Author estimated such ratio to be 15% and mentioned that the value is consistent with the values found by (Malamud et al. 2004). In the mentioned paper, three different values are obtained in three different areas of the world, namely 0.24%, 0.3% and 0.6%, none of them is any near to the 15% calculated here. In my opinion, there is no reason why such numbers should be comparable, but still the claim seems to be wrong, if I am not mistaking. There are other examples in which the Author refers to literature in a sloppy way. For example, he mentions Voight model and inverse velocity method without spending a line explaining what they are; in Section 1.2.1 it is mentioned that "the inverse velocity method is embedded in a Bayesian DSS", without explaining what it even means. In the following line, he claims that (Manconi and Giordan 2015) stated the Se=75% "can be considered as a high reliable value" while, in the paper he refeers to, there is no reference whatsoever to Bayes theorem or to Se. In their paper, 75% is a "measure of reliability" given by a "normalized Pearson's coefficient between model and data": what is the relationship with Se, if any?

In conclusion, I believe that the proposed Manuscript is not suitable for publication in NHESS, both for the organization of the paper, the method used, and the conclusions (or lack thereof) drawn.

---

## Author Comment (AC2) · 5 Mar 2018

By: Davide Bertolo

The Manuscript deals with the issue of early warning systems for active slow moving landslides. A method is proposed for a "decision support system", claimed to be objective and reliable, which was applied to "some" (three) cases in the Italian Alps.

AC: As a matter of fact "some" is more than two, anyway I'll look for a more elegant word. Anyway, as required by Ref. 1, the chapters regarding the two other cases will be suppressed and will be only cited.

I believe that the Manuscript is written in a rather confuse way, and that the above claims are not supported by the actual content of the text. Morevoer, it occasionally contains questionable statements.

Overall review of the Manuscript: The Abstract of a scientific paper is supposed to concisely describe the framework, purpose, method, results and conclusions; the proposed Abstract hardly contains such information.

AC: I share your advice about the quality of the abstract. The abstract is not adequately concise. It will be improved according to the suggestions of the referees.

Then, from the beginning of the main text up to section 1.2, the Manuscript is written in a colloquial fashion and it does not introduce the general framework of the problem to put the proposed method into proper context.

AC: Sorry but I can't agree with your above statement. The reasons that fostered us to develop a DSS able to match quantitative and qualitative data are based on a field experience which has been described in detail.

This is an experience that has been made during a real emergency (see section 1.1 of the manuscript). On the other hand, I agree that the exposition of the general framework is, if anything, too detailed and it is too wordy. Anyway, since your suggestions about improving the conciseness of the manuscript are welcome, it will be reduced as much as possible, aiming to make the whole paper clearer.

The description of the "decision support system" is given in a qualitative way, without motivating the adopted numerical values or, worse, by explicitly mentioning that values are "reasonable", without providing any insight into the consequences of using those particular values, or uncertainty, or any form of validation. At least these are not explained.

AC: "explicitly mentioning that values are "reasonable"". The word "reasonable" has not been used except for the displacement of rock volumes (line 39, pag.11). It's worth to emphasize that, when monitoring unstable bulk rocky masses, a LOS displacement rate of 5mm/h is already a consistent one, regardless to the volume. This is an empiric displacement threshold and the word "reasonable" was intended to mean a conservative threshold. The word will be suppressed.

It's my understanding that, perhaps, you got the impression that a sort of "rule of thumbs" had been implemented (I apologize for the colloquial tone) using values that have been arbitrarily adopted.

Nevertheless, I have to thank you again for having stressed one of the critical topics of the Bayesian method described in the manuscript: that is setting correct values for Pr, Se, Sp.

The most difficult value to set is that of the Prevalence, Pr, because it strongly influences the PPV of the decisional steps. It's evident that a value of Pr of 0,15 is leading to a PPV less than 50%.

Paradoxically, the above results show that, to issue a civil protection alert, it would be a more reliable method to toss a coin (PPV=50%) than to adopt the threshold values provided by a single monitoring network when not managed by a comprehensive decision support system.

The period "Moreover, for which regards the assessment of Pr and Sp values, it can be noticed that an accurately calibration of their values It's not so relevant to improve the reliability of a DSS, even though that is always a good achievement." is misleading because it's poorly and mistakenly phrased. The reader gets the impression that it's not worth to search for more accurate values of Pr, Se and Sp, while the intention is to show that makes more sense to search for elements supporting a decision than to increase the accuracy of these parameters.

The two following sections are supposedly devoted to the application of the same method in two different sites. It is not clear to me what is it that the Author learned from the first case, let alone that the information provided in the two additional sections does not provide any more insight. Thus, how can the Author state, in the first few lines of the Conclusions section, that the method was "successfully applied"?

AC: As stated before, the two sections will be suppressed. The two other sites show different geological situations, if compared to the Mont de La Saxe. In these two cases the expected events are, respectively, a rockslide and a rockfall. Rockslides and even more rockfalls usually occurs too rapidly and the use of quantitative EW systems is almost useless, if these are used in a standalone configuration. That's the reason why, under a conservative approach, other qualitative clues can be integrated with the quantitative clues coming from the monitoring systems are not sufficient to support a civil protection alert or can be detected too late.

The early warning monitoring system activated at the Mont de la Saxe landslide in 2009 was initially based on instruments providing only quantitative data, and the issuing of a civil protection alert was supposed to be issued immediately at the exceeding of the displacement thresholds.

The rigorous application of that procedure could certainly lead to the issuing of several false alerts and to the unmotivated closing of the A5 motorway. The A5 motorway is one of the most important routes between Italy and France. It's my fault and I apologize not to have highlighted that the closing of that route causes to the Italian economy a damage equivalent to EUR 1.000.000 for every day of closing (bibliographic references will be added), not counting the damages to the French economy. Under that light it becomes clear that the "successful application" of the method consists in having implemented a DSS that lead to the application of civil protection measures only when strictly required and unavoidable, saving a great amount of financial resources.

Where are evidences supporting that claim? The remaining of the Conclusions section fails to report actual "conclusions" of the effort made by the Author. What do we learn from reading the Manuscript?

AC.: The conclusions are widely exposed in section 6. The paper exposes the results coming from an operational experience where a new decisional procedure method had to be adopted and experimented directly "on the job", to prevent the issuing of unjustified alerts. The conclusions are, in my opinion, clearly exposed and one of the most important lessons learned is that decision support systems for critical landslides must provide probabilistic forecasts where the probability must be evaluated by a transparent procedure. This procedure has been exposed in the manuscript and, obviously, can be improved.

Why does the Author suggests not "installing useless and redundant instrumentation", where is evidence supporting that statement?

AC: The procedure proposed has been codified in a transparent decision support system, the decisional value require data that, in each case can be provided only by specific monitoring networks or field observations, this means that it helps the designing of integrated early warning systems avoiding the installation of the networks that don't provide data suitable to be inserted in the DSS.

The reason why would he like to revise the commonly accepted definition of landslides is also obscure to me.

AC: It would be presumptuous of me to propose the "revision of the commonly accepted definition of landslides". As a matter of fact, according to the actual text of the manuscript, it's explicitly suggested to adopt existing definitions and classifications of landslides like the one by Cruden and Varnes (see p. 16, row 26).

Additionally, the paper suggests the adoption of classification criteria based on the displacement velocity only when adopting the proposed DSS, to increase the accuracy of the Pr value. This proposal is made because a classification of landslides in a given area, based on the terrain displacement velocity, is objective and replicable and, at a regional scale (e.g. using PS Journals) allows to perform an accurate screening of the active landslides.

Eventually, the language throughout the text is sloppy; I am not an English native speaker myself, but I could spot several mistakes and, occasionally, repeated words and even a three-lines sentence (end of page 6) denoting careless writing.

AC: as previously said, English will be furtherly improved.

Additional comments:

There are many expressions used in a confusing way. For example, the Author refers to "numerical methods" for landslide monitoring: "numerical methods" are usually devoted to computer-based problem solving, typically optimization or differential equation solving, which is not the case here.

AC.: The monitoring systems that measure the Mont de la Saxe landslide displacements are based on robotized total stations (RTS), Ground Based InSAR and GPS. These instrumental networks produce quantitative data which are processed by expert systems to calculate mean displacement values and to generate automatic alerts based on quantitative thresholds. Therefore, these are numerical systems according to your interpretation too.

Please refer to:
- SALVANESCHI, P.; CADEI, M., LAZZARI, M. (1996). "Applying AI to structural safety monitoring and evaluation". *IEEE Expert - Intelligent Systems*. **11** (4): 24–34. doi:10.1109/64.511774.
- LAZZARI M. AND SALVANESCHI P. (1999): "Embedding a geographic information system in a decision support system for landslide hazard monitoring". International Journal of Natural Hazards, 20 (2-3), 185–195.
- BARONI F. AND LAZZARI M.: "Expert systems as tool for adult education: A qualitative research twenty years after". Mondo Digitale, Volume 13, Issue 51, 1 June 2014, Pages 801-810.

In these papers is cited the Eydenet system, used by the Regione Autonoma Valle d'Aosta geological survey to elaborate numerical data from its landslide monitoring systems,.

There are whole sentences that have little to do with the content of the paper (i.e. lines 29-38, page 2, about "human behaviour" and its supposedly relevant role in decisional processes),

AC: It has been exposed in the manuscript introduction that the aim of the DSS is to provide a decisional path able to combine all the data available to the decision maker at a given time, during the landslide evolution. As described in the Mont de La Saxe case, landslide early warning procedures are usually based on final decisions made by human beings. The proposed DSS is a protocol designed to steer the decision maker to the best decision that is possible to make at a given time with all the evidences available, minimizing the human biases which can influence the decision maker judgement.

or the many references to "clinical practice", whose only overlap with the content of the Manuscript is the use of Bayes theorem.

AC: The reasons for an extensive reference to the medical literature have been widely exposed in the text (e.g.: line 30, page 5; line 3 and remainder, pag. 6). Decisional protocols based on the Bayesian iteration form the basis of clinical protocols and are widely adopted in the clinical diagnosis. Therefore, it's not far-fetched to look to the clinical experience when dealing with complex systems, like slope instabilities, whose behaviour is non-linear, thus difficult to describe and to predict when an exclusively deterministic approach is adopted.

There are a few confusing sentences bringing conceptually different issues to the same level, for example: i) in line 39, the Author refers to "monitoring" and "bias" on the same footing, while monitoring does not contain biases per se, being a quantitative measure; it may have different degrees of approximations or confidence, but surely it does not represent a bias;

AC: I assume that you refer to line 39, pag. 15. Maybe that the "bias" word is misleading in that context, and it will be replaced to make the text clearer. It is sure that it doesn't reflect my opinion about the quantitative measures. I agree with you that quantitative measures can't be biased as it is stated at line 8, pag. 2.

ii) in line 6, page 3, the sentence "Such evidences, which are the so-called expert judgements" contains the same kind of flaw: an "evidence" cannot be a "judgement".

AC: in operational geology, terrain observations and their interpretations are always affected by a certain degree of subjectivity. In that light, a terrain observation, such as the appearance of a tension crack is an "evidence", its interpretation as a forerunner of an imminent collapse is a "judgement" based on the experience of the observer, too. That's the reason why these kinds of observations are named in the paper "expert judgements". This definition comes from the studies made by KAHNEMAN (refer to bibliography).

Similar and even worse problems are represented by the use of the word "objective" throughout the Manuscript. Objectively defining values for parameters in a model, such as the various thresholds and percentages values used here, is not equivalent to select them arbitrarily.

Of course one can follow a trial and error procedure, but in that case one would also have to make sure if initail (the trial part) values were properly selected, or should they be modified (the error part). It seems that the decisions system's parameters were rather arbitrarily selected; if not, the Author failed to describe the method properly, in my opinion.

AC: the objectiveness of the method regards its structure and the clear decisional path that is followed to come to a final decision that could have negative effects on the civil protection system's reliability and producing relevant economic damages like the ones avoided to the Italian Economy in 2013 and 2014.

Of course, the values of the parameters adopted may be questionable, and it is one of the key issues to deal with, applying the proposed DSS to a specific landside.

In addition, I would like to point out that the values of the parameters have been not "arbitrarily" selected. Indeed, a relevant part of the manuscript has been devoted to the issue of how to correctly setting values of Pe, Pr, Se.

It has been extensively highlighted (line 3, Pag. 9, and section 6) that the values of some parameters are influenced by the robustness of the landslides inventories which can be decisive in assessing a consistent value of Pr.

On the other hand, regarding Pe, its value is provided by an analysis of the landsides actually collapsed after having exceeded a certain displacement velocity. This topic is widely discussed in the paper, focusing on the need to obtain reliable statistic records (how many landslides, that exceeded the displacement thresholds, collapsed?). The real weakness of the thresholds calculated using the Voight method based on statistical analyses is due to the too short historical record that support their calculations. Moreover, the population of the landslides on which the thresholds are calculated almost never includes the landslides that actually didn't collapse even though exceeding the displacement thresholds (i.e. the false positives), which abruptly increases the specificity (Sp) of the test and the risk of issuing a false alert based exclusively on the displacement thresholds.

The trial and error method that you suggested is a useful way to increase the reliability of some parameters used in the DSS and to increase the accuracy of their values and, in my opinion, that is another reason to adopt the proposed DSS.

And, by the way, "parameters" are contstant values in a mathematical model, which are used to represent the relationship between the "variables" of the model itself; parameters can be tuned, not measured (at variance with line 5, page 10). Variables are the measurable quantities, and bear physical meaning.

AC : definition from the Oxford dictionary of English:

"PARAMETER: 1 *technical* : A numerical or other measurable factor forming one of a set that defines a system or sets the conditions of its operation." This definition of parameter has been adopted.

The use of Bayes theorem is actually my major concern, here. Bayesian inference is a statistical method, i.e. it is meant to estimate a probability density function, based on the concepts of prior and conditional probabilities, and on the idea that posterior probabilty can be updated whenever new evidence comes about. It is understood that new evidence must concern the SAME variables used in previous estimates. Otherwise, we start describing a different phenomenon, or a different model. It seems to me that in the process described by the Author each different step (cf. Fig. 5) brings about NEW variables, and no probability is defined for the overall set of variables to simultaneously assume different values (conditional probability), and no prior estimate of such variables to assume those values was defined (prior probability). If so, I do not see how can one claim to enforce Bayes theorem, or I am missing something fundamental here. This add to the issue of how the numerical values of the thresholds were selected, discussed above.

AC: Bayesian inference is a well codified process to support decision-making processes. The process consists in testing the consistency of a given hypothesis (in this case a scenario of collapse) looking for elements that, if present, can support the original hypothesis increasing its likelihood. If those elements are not present, the original hypothesis is not sufficiently consistent and other scenarios are equally likely to happen.

Moreover, in line 32, page 7, what does "Applying the corrections required .. [1] becomes" mean? Correction required by whom? How is Eq. (2) implied by Eq. (1)?

AC: The Bayes theorem, as exposed in the (1), is its generic formulation. In the case of Bayesian inference the adoption of Pe, Se, Sp and their insertion in the Bayes general formula must be intended as a "correction", i.e. an adjustement, of the formula for its specific application to the case. The correction is not

to be intended as the correction of a mistake but as the application of the correction factors. The latter term "correction factors" will be adopted in the reviewed manuscript, since it seems to be clearer.

In line 11, page 8, the Author refers to susceptivity (usually referred to as susceptibility) as the ratio of unstable area to total area under investigation, which is a bit of a semplification, given the body of literature existing on the topic of landslide susceptibility assessment.

AC: as before stated, the susceptivity is Se, intended as the percentage of a given area affected by landslides observed at a specific time. The aim of the manuscript is not to deal with the topic of the landslide susceptibility assessment.

Also, the Author estimated such ratio to be 15% and mentioned that the value is consistent with the values found by (Malamud et al. 2004). In the mentioned paper, three different values are obtained in three different areas of the world, namely 0.24%, 0.3% and 0.6%, none of them is any near to the 15% calculated here.

AC:  The paper by Malamud et al. (2004) was mistakenly cited and erroneously and unintentionally pasted. It should have been removed before the submission.   My apologies to the Editor and to the reviewers.

 In my opinion, there is no reason why such numbers should be comparable, but still the claim seems to be wrong, if I am not mistaking. There are other examples in which the Author refers to literature in a sloppy way. For example, he mentions Voight model and inverse velocity method without spending a line explaining what they are;

AC: Due to length of the manuscript, it seemed to me useless to explain methods which should be very well known in the field of landslide monitoring. Both Voight and the Fukuzono methods are widely applied and well known.

in Section 1.2.1 it is mentioned that "the inverse velocity method is embedded in a Bayesian DSS", without explaining what it even means. In the following line, he claims that (Manconi and Giordan 2015) stated the Se=75% "can be considered as a high reliable value" while, in the paper he refeers to, there is no reference whatsoever to Bayes theorem or to Se. In their paper, 75% is a "measure of reliability" given by a "normalized Pearson's coefficient between model and data": what is the relationship with Se, if any?

AC: It was extensively explained in the paper that the DSS is a system which allows the decision-maker to decide whether to issue a civil protection alert or not, This decision is supported by gathering all the available clues that are provided both by data from instrumental networks and terrain evidences. As explained in the paper, during the Mont de La Saxe emergency state (2014), we adopted in-parallel both the Voight method, which was the original method used to set the early warning thresholds, and the Inverse-velocity method (the results of that test were later exposed in the Manconi and Giordan paper cited in the manuscript).

The term "embedding" refers to using the Time of Failure (or ToF) provided by the inverse velocity method as one of the elements provided by the quantitative systems to support the decisions made by the decision maker at a given step of the decisional process.

My apologies for the mismatch. The 75% value, calculated by the two Authors, is the reliability of the forecast at a given time and refers to the value actually to be assumed for Sp and not, as in the paper, to

Se. It can be used in the decisional process as Sp, that is the probability that the used test can detect a "true positive". It is going to be corrected.

Sincerely,

Davide Bertolo

05.03.2018